# Position: Explainable AI Cannot Advance Without Better User Studies

**Matej Pičulin** [1]   **Bernarda Petek** [1]   **Irena Ograjenšek** [1]   **Erik Štrumbelj** [1]

## Abstract

In this position paper, we argue that user studies are key to understanding the value of explainable AI methods, because the end goal of explainable AI is to satisfy societal desiderata. We also argue that the current state of user studies is detrimental to the advancement of the field. We support this argument with a review of general and explainable AI-specific challenges, as well as an analysis of 607 explainable AI papers featuring user studies. We demonstrate how most user studies lack reproducibility, discussion of limitations, comparison with a baseline, or placebo explanations and are of low fidelity to real-world users and application context. This, combined with an overreliance on functional evaluation, results in a lack of understanding of the value explainable AI methods, which hinders the progress of the field. To address this issue, we call for higher methodological standards for user studies, greater appreciation of high-quality user studies in the AI community, and reduced reliance on functional evaluation.

## 1. Introduction

Explainable artificial intelligence (XAI) plays an important role in artificial intelligence (AI), machine learning (ML), and other areas of quantitative data analysis. As these technologies become more integral to our professional and personal lives and face greater legislative oversight, the importance of XAI continues to grow.

Academic interest in XAI is also growing rapidly. On Dec 29, 2024 there were, according to Scopus, 14077 works with at least one of the terms *explainable AI*, *XAI*, or *explainable artificial intelligence* in the title, abstract, or keywords. More than half of these (8,945) are dated 2024. There are

also numerous survey papers that look at XAI from different angles and domains, including a survey of surveys by Schwalbe & Finzel (2023).

We take a closer look at human subject-based evaluation of XAI - evaluating XAI by studying the performance, behavior, and opinions of human subjects (*user studies* for short). In particular, we focus on the quality and scope of user studies and results that have implications for future user studies and the field of XAI as a whole. We do not discuss parts of XAI that are extraneous to user studies and their role in evaluating XAI. For readers interested in these, we recommend Schwalbe & Finzel (2023) as a starting point.

We aim to establish two points:

- **User studies are key to evaluating XAI.** The user is an integral part of XAI and if our goal is to understand the value of XAI methods, there is no alternative to user studies.

- **The current state of user studies in XAI is poor.** There is a clear lack of quality in all aspects of user study design, from defining the purpose of the study and participant selection, to study methodology and task design. In fact, most user studies in XAI are below the threshold of what is acceptable in fields of science with a longer history of human subject-based research. Also, most user studies are conducted with very low fidelity to real-world use.

If we accept both that user studies are key to understanding the value of XAI methods and that the state of user studies is poor, then it follows that our current understanding of the value of XAI methods is poor. And the only way to move forward is to substantially improve at least the quality, but hopefully also the quantity of user studies.

**It is our position that we must change our mindset about user studies. Well-designed user studies should be encouraged, because at this point their contribution to XAI is more important than most further theoretical or non user-centric empirical advancement. On the other hand, user studies for the sake of doing a user study or as a methodological afterthought, should be discouraged, as they contribute little to our understanding of XAI.**

[1]Faculty of Computer and Information Science, University of Ljubljana, Večna Pot 113, Ljubljana, Slovenia. Correspondence to: Erik Štrumbelj <erik.strumbelj@fri.uni-lj.si>.

*Proceedings of the 42nd International Conference on Machine Learning*, Vancouver, Canada. PMLR 267, 2025. Copyright 2025 by the author(s).

## 1.1. Related Work

Evaluating XAI is an active area of research, with several works in part or fully dedicated to user studies. We draw heavily from the seminal work of Doshi-Velez & Kim (2017; 2018), the comprehensive survey by Nauta et al. (2023), and the taxonomies and classifications of evaluation and user studies in XAI (Chromik & Schuessler, 2020; Herm et al., 2022; Lai et al., 2023; Lopes et al., 2022; Rong et al., 2023).

Unlike related work, we focus on the quality of user studies, their fidelity to real-world use, and the implications for our understanding of XAI. Our main contribution is our position, supported by an analysis and synthesis of related work and an analysis of 607 XAI papers that feature a user study.

## 2. The Role of User Studies in Evaluating XAI

We adopt the perspective of Speith & Langer (2023) that all XAI eventually aims to satisfy societal desiderata, such as trust, fairness, or downstream task performance. Speith & Langer (2023) propose the following model: XAI provides explanatory information, which facilitates understanding, which in turn affects how well the societal desiderata are satisfied. Note that Lopes et al. (2022) also propose a similar but more detailed model of human-centered evaluation.

The satisfaction of societal desiderata provides feedback on the appropriateness of the explainability approach. Speith & Langer (2023) divide evaluation methods into three categories: *explanatory information*, *understanding*, and *desiderata* evaluation methods. Explanatory information methods are concerned with how accurately XAI describe the AI system (for example, fidelity and completeness). Understanding methods are concerned with how well the XAI facilitates understanding of the AI system (for example, subjective understanding or being able to predict model behavior).

Both explanatory information and understanding evaluation methods allow for evaluation without human subjects (or *functional evaluation*, as it is referred to in the popular taxonomy by Doshi-Velez & Kim (2017; 2018)). However, evaluating societal desiderata requires human subjects.

The view that satisfying societal desiderata is the end goal and that explanatory information and understanding are just proxies, is generally accepted: Adadi & Berrada (2018) state four reasons for XAI: to justify, to control, to improve, and to discover. Doshi-Velez & Kim (2017; 2018) state that interpretability is often used as a proxy for other criteria, such as fairness, safety, and trust. They also raise the question of downstream goals of interpretable ML systems and why interpretability is the right tool for achieving those goals. (Vilone & Longo, 2021) state that the construct of explainability is linked with other constructs such as trust,

transparency, and privacy. (Lipton, 2018) state for post-hoc interpretability, that work in this field should fix a clear objective and demonstrate evidence that the offered form of interpretation achieves it.

If societal desiderata are the end goal and cannot be evaluated without humans, then user studies are an important part of XAI evaluation. Or, as we and others argue, they are essential (Buçinca et al., 2020; Doshi-Velez & Kim, 2017; 2018; Vilone & Longo, 2021; Zhou et al., 2021).

### 2.1. Misalignment Between Academia and Practice

Poorly designed user studies, or the absence of user studies altogether, result in a lack of understanding of what works and what does not work in practice. This understanding is also important for guiding theoretical and methodological developments. Without it, there is likely to be a misalignment between academia and practice.

Indeed, this is not only our view, but also an increasingly common theme in related work. Bhatt et al. (2020) emphasize that there is a gap between XAI research and what is needed in practice. Decker et al. (2023) state that the academic XAI toolbox is not fully utilized in practice and practitioners call for tools that do not yet exist. Ghassemi et al. (2021) argue that current XAI methods are unlikely to achieve transparency and mitigate bias in healthcare. Kong et al. (2024) state that human-centered XAI may still lack explicit guidance of methods developing explainability solutions for different stakeholders. Preece et al. (2018) argue that failure to satisfy users of AI technology in the long run will be the most likely cause of another AI Winter. Lai et al. (2023) argue that the focus and design of studies may not align with how AI is or will be used in real-world decision-support applications. And Lopes et al. (2022) state that there is still a clear disconnect between technical XAI approaches and their effectiveness in supporting users' objectives.

## 3. Challenges in XAI User Studies

In this section we survey and discuss the general sentiment and specific issues that are relevant for designing user studies in XAI.

Several authors point to *a lack of formalism and consensus in terminology* (Jung et al., 2023; Lai et al., 2023; Lopes et al., 2022; Markus et al., 2021; Zhou et al., 2021) and call for *more standardized evaluation and reporting methodologies* (Sperrle et al., 2021). This can be attributed, at least in part, to XAI being a young field. However, standardized user study methodologies can be drawn from fields with longer traditions in such research, such as psychology and HCI. Moreover, there have been developments in formal frameworks for explanation (Adolfi et al., 2025; Bassan et al., 2024; Barceló et al., 2020; Vilas et al., 2024).

Several authors call for *more and better user studies* (Buçinca et al., 2020; Gurrapu et al., 2023; Jacovi et al., 2021; Johs et al., 2022; Keane et al., 2021; Zhou et al., 2021).There is a consensus that *evaluating XAI is an interdisciplinary effort* (Lopes et al., 2022; Zhou et al., 2021) and that there is a lack of multidisciplinarity in papers (Lopes et al., 2022). Authors argue that the AI/ML community should draw from human-computer interaction (HCI) (Alangari et al., 2023; Lai et al., 2023; Liao & Vaughan, 2024; Williams, 2021), the Human-Human trust community (Vereschak et al., 2021), or social and behavioral sciences (Alangari et al., 2023; Johs et al., 2022; Miller et al., 2017; Miller, 2019). Some authors go further and argue that the HCI community should be the driving force (Chromik & Schuessler, 2020; Vilone & Longo, 2021).

*Interactive explanations* are also starting to receive more attention. See Bertrand et al. (2023) for a review, Boukhelifa et al. (2018) for evaluation of interactive machine learning systems, and Chromik & Butz (2021) for a review of interactive explanation user interfaces. Nguyen et al. (2024) argue that interactivity is key for real explainability and actionable understanding. Abdul et al. (2018) and Williams (2021) argue that interactive explanations should be explored further. However, most XAI methods and even more user studies, are static (Abdul et al., 2018).

Several authors point to *the importance of taking into account user's mental models and cognitive processes* (Hoffman et al., 2018; Kenny et al., 2021; Lopes et al., 2022; Rong et al., 2023), but there are still few user studies that do so.

### 3.1. A Diverse Range of Stakeholders

Applications of XAI involve a diverse range of stakeholders, including developers, researchers, end-users, decision-makers, regulators, educators, and policymakers. However, much of academia's focus is on the AI/ML practitioner and iterating between model development and evaluation.

As a result, key stakeholders, particularly end-users, are underrepresented in the literature. The study design choices in current research often fail to align with real-world decision-support applications. For example, tasks based on readily available datasets may not reflect realistic decision-making scenarios. Healthcare might be an exception, as many user studies focus on end-users (Jung et al., 2023).

Ideally, as several authors have also pointed out, proper evaluation of XAI would involve identifying and engaging all stakeholders while understanding the role of XAI in the context of use, domain, and end-users' expertise (Bhatt et al., 2020; Decker et al., 2023; Kong et al., 2024; Lai et al., 2023; Langer et al., 2021; Lopes et al., 2022; Nguyen & Zhu, 2022; Preece et al., 2018).

### 3.2. Personalized XAI

There is growing evidence that personal characteristics influence how people perceive and interact with XAI. Buçinca et al. (2021) suggest that human cognitive motivation moderates the effectiveness of explainable AI solutions. Reeder et al. (2023) find differences in trust and understanding based on gender and educational background. Millecamp et al. (2019) find that personal characteristics have significant influence in recommender systems, and that this influence is moderated by explanations.

Subsequently, more and more authors advocate for the use of user-centered methods, developing XAI with the end-user in mind, and tailoring XAI to different end-users (Rong et al., 2023; Ribera & Lapedriza, 2019; González-Alday et al., 2023). However, few user studies explore inter-personal differences. Anjomshoae et al. (2019) also find in their survey of explainable agents that only a few works addressed the issues of personalization and context-awareness.

### 3.3. Links between Functional, Perceived, Proxy, and Real-World Results

Ideally, a user study would evaluate the target XAI method in a real-world context, but such studies are the most challenging to conduct. Evaluation becomes easier with proxy tasks and even easier when relying solely on self-reported percieved quality. Evaluation without users (functional evaluation) is the simplest, but least impactful.

However, validating an easier evaluation method in a more complex context allows us to retain the benefits of simplicity without sacrificing impact. In this section, we summarize empirical findings on such relationships.

Notably, we found no studies exploring the relationship between functional evaluation and user performance. However, we did find several studies that explore other types of relationships:

Hase & Bansal (2020) found that subjective user ratings of explanation quality are not predictive of explanation effectiveness in simulation tests.

Amarasinghe et al. (2024) show the importance of closely reflecting the deployment context, by demonstrating that there is no practical utility of explanations. They also find a mismatch between self-reported metrics and improvement in decision-making.

Buçinca et al. (2020) performed three experiments to compare using proxy tasks and using subjective measures of trust and preference as predictors of actual performance. They found that proxy tasks did not predict the results of the actual decision-making tasks. They also found that subjective measures did not predict objective performance.

Chromik et al. (2021) investigated how non-technical users form their mental models of global AI model behavior from local explanations and found that participants overestimated their understanding.

In summary, we know little about the relationships between these levels of evaluation. What we do know suggests that context plays a crucial role and that designing functional evaluation methods or proxy tasks to predict real-world performance may be challenging.

### 3.4. The Placebo Effect and Placebic Explanations

The placebo effect is well known in medicine, where the effect of a new treatment is compared to a placebo control, such as a sugar pill or saline injection. The goal is to measure the effect of the treatment itself, beyond the impact of its administration. This principle is also applied in other fields, such as psychology, sports, visualization research (Kosch et al., 2023), and marketing (Vaccaro et al., 2018).

Kosch et al. (2023) identified three types of HCI user studies:

- A *conventional user study* (compare a novel system with a baseline),

- a *placebo-controlled study* (compare a novel system with a system that pretends to have a novel functionality), and

- a *placebo study* (compare a baseline system to a system that pretends to have a novel functionality).

Most XAI user studies are conventional, where the baseline is an existing explanation or no explanation. The positive results of such studies are questionable, as they could stem from a novel explanation or the placebo effect. This is particularly problematic when measuring trust, perceived performance, or other subjective aspects.

Recent research has highlighted this issue and advocates for placebo-controlled studies in XAI (Eiband et al., 2019; Bosch et al., 2024). However, conducting such studies in XAI requires placebic explanations. Unlike in medicine, where the treatment can be easily decoupled from its content, this is a greater challenge in XAI.

A few examples of placebic explanations include Liu (2021), who used tautological statements, which work well with textual explanations, but this cannot be generalized since XAI explanations take many forms. Textual variations were also used by Pias et al. (2024) and Eiband et al. (2019), while Wang & Ding (2024) created placebic explanations for feature importance by randomly shuffling contributions. Similarly, (Kenny et al., 2023) created placebic explanations

in reinforcement learning by randomly perturbing prototype images, thus disassociating them with intuitive actions.

Some placebo studies use *no explanation* as a baseline. For example, Eiband et al. (2019) conducted a no explanation/placebo/real explanation study. They found that placebo explanations invoke levels of perceived trust similar to real explanations. Kosch et al. (2023) and Villa et al. (2023) also show that subjective measurements improve, while objective measurements remain unchanged when the system is described as having AI. Kloft et al. (2024) suggests this effect is not only due to verbal descriptions but also the socio-technical context. Note that these studies used sham systems and did not consider objective metrics like accuracy or time. We did not find an XAI user study presenting a novel explanation with a placebo-control group.

### 3.5. Concerns with Crowdsourcing Platforms

Crowdsourcing platforms (Amazon Mechanical Turk, Prolific, CloudResearch, etc.) have become very popular in recent years, primarily because of their convenience. XAI is no exception (see Section 4).

As the popularity of crowdsourcing increased, questions about the demographics and data quality of the crowdsourcing samples compared to other samples and the general population have emerged. The demographics and personalities of Amazon Mechanical Turk workers (MTurkers) consistently differ from other samples and the general population (Burnham et al., 2018; Douglas et al., 2023; Goodman & Paolacci, 2017; Paolacci & Chandler, 2014; Weigold & Weigold, 2022). For example, MTurkers are more educated than the general population and older MTurkers may have higher cognitive abilities than the corresponding age group in the general population (Ogletree & Katz, 2021).

MTurkers may be less attentive than students (Aruguete et al., 2019; Barends & De Vries, 2019; Goodman et al., 2013; Tahaei & Vaniea, 2022) when completing tasks. Inattentiveness can also manifest as inconsistencies in answers that don't make sense (Kay, 2024). MTurkers are very much prone to multitasking (Brigden, 2024; Necka et al., 2016). Moreover, recent studies suggest that they often engage in satisficing and low-effort behaviors, potentially compromising previously validated study results (Berry & Burton, 2024; Berry et al., 2024). Additionally, MTurkers can easily misrepresent themselves to qualify for specific studies, posing a significant concern for researchers who require participants with particular traits (Ahler et al., 2021; Dennis et al., 2020; MacInnis et al., 2020; Moss et al., 2021). Besides being dishonest, another issue with screening procedures that include subjective surveys is the potential overconfidence of MTurkers (Tahaei & Vaniea, 2022).

Moreover, a small sample of MTurkers tends to be highly

productive and consequently very familiar with the studies they participate in. This lack of naivety can impact data quality, especially if similar attention checks are frequently used (Chandler et al., 2014; 2015; Necka et al., 2016; Stewart et al., 2017). Another potential issue is the interaction among MTurkers on forums, which can influence the participant pool by preferring some researchers over others (Chandler et al., 2014). Additionally, high attrition rates could be a problem, as MTurk workers can leave studies with just a click (Arechar et al., 2018; Zhou & Fishbach, 2016).

Most importantly, with the growing population of MTurk, the quality of the data seems to have decreased in studies that have been recreated or reviewed over time (Chmielewski & Kucker, 2020; Kennedy et al., 2020; Marshall et al., 2023). Consequently, researchers are starting to focus on alternative crowdsourcing platforms and survey platforms that offer online sample and panel services. It looks like MTurk provides lower quality data compared to Prolific, CloudResearch (Connect), and Qualtrics (Qualtrics Panels) (Albert & Smilek, 2023; Douglas et al., 2023; Eyal et al., 2021; Peer et al., 2023). However, little research has been done yet on how these other platforms compare to other population samples or the general population. There may be some potential data quality issues with these platforms compared to student samples (Novielli et al., 2023; Tahaei & Vaniea, 2022).

These findings at best further emphasize the importance of carefully designing a crowdsourcing user study and at worst put into question results obtained via crowdsourcing.

## 4. The State of User Studies in XAI

We analyzed user studies from 607 XAI academic papers. Throughout, we grouped the papers into papers published *up to 2020*, papers published *2021 and after*, and papers published in *top 4* conference venues (NeurIPS, ICLR, ICML, AAAI) from 2020 and after. Every paper from the top 4 group is also in one of the former two groups. Note that some results are on all papers, while others use a random subsample from each group. Details of how we collected and analyzed the papers can be found in Appendix A.

### 4.1. Publication Year

The distribution of the papers over publication year is shown in Figure 1. The rising popularity of XAI in academic research is clear. The lower paper count in 2024 can be explained by the fact that most of the papers were collected in early 2024. However, we have most likely collected proportionately fewer papers in the most recent years, because the sample is biased towards papers published earlier (see Appendix A.3 for a discussion of the limitations).

A more thorough search for relevant papers in the top 4 venues doubled the number of papers for those venues in that period. From this we can estimate that the 607 papers represent at best one half of the total number of XAI papers with a user study in the 2020-2024 period (possibly more prior to 2020).

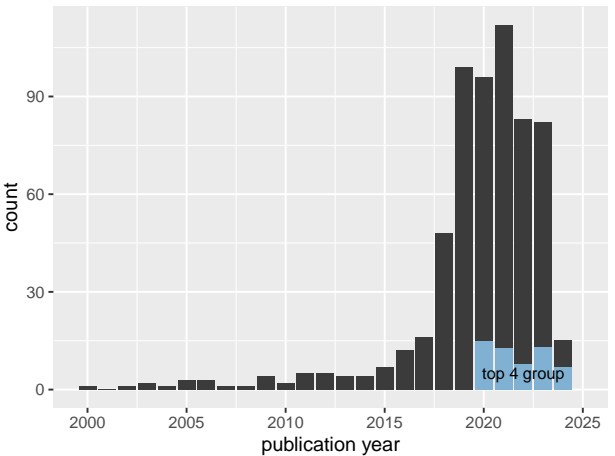

*Figure 1.* Distribution of papers over publication year. Papers from the top 4 group are highlighted.

### 4.2. Participant Count and Type

Figure 2 shows the distribution of user study participant count. The distribution is similar for all three groups. Most studies are under 50 participants, with a long tail of studies with more than 50 participants. These roughly correspond to non-crowdsourcing and crowdsourcing studies, respectively.

Figure 3 shows the distribution of user study participant type. The most common approach is to use a crowdsourcing platform, typically Amazon MTurk. The second most common approach is to recruit students, sometimes combined with university researchers and administrative staff. These approaches represent more than two thirds of all studies. Between 20% and 30% of studies are done on domain experts and probably less than 10% on AI experts. Studies from the top 4 group lean more towards crowdsourcing (or not stating particpant type at all) and less frequently include other types of participants, in particular, domain experts. A non-negligible proportion of studies (10% - 20%) do not state participant type.

### 4.3. Study Design and Quality

The estimates of the variables that measure the methodological quality of user studies are summarized in Figure 4. The standout result is that very few user studies from the top 4 group contain enough information to be considered

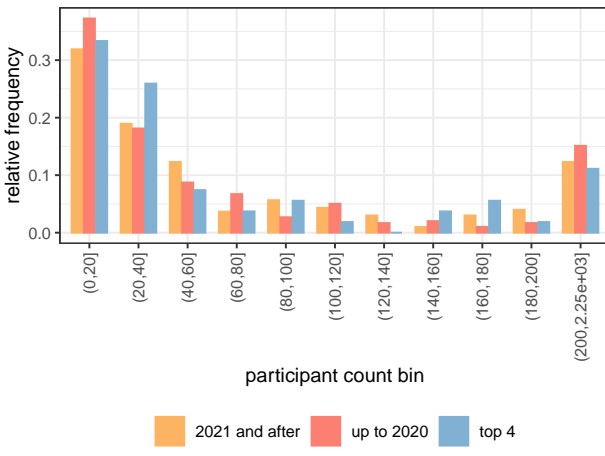

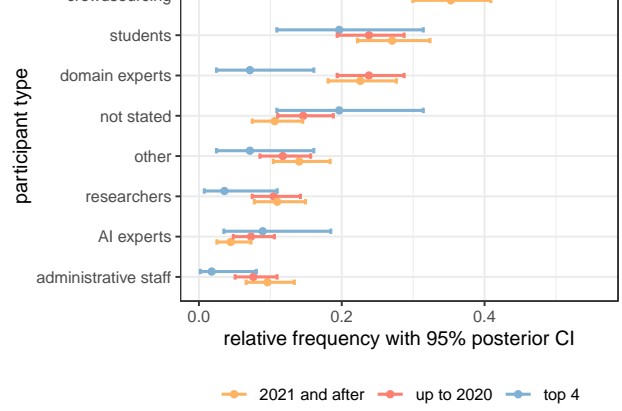

*Figure 2.* Distribution of user studies over participant count bins. Note that there were a total of 647 user studies in the 607 papers, 48 of which (7.4%) did not report participant count.

*Figure 3.* Relative frequencies user study participant type. Note that a user study can have multiple participant types, so a group's relative frequencies might not sum to 1.

reproducible. This is in stark contrast with the rest of the papers, where we estimate between 50% and 90% are reproducible. There also appears to be some improvement for the more recent papers in reproducibility, stating the limitations of the user study, pre-testing, and attention checks in crowdsourcing studies.

Other results are similar for all three groups. Preregistration and pretesting are rare. Validated questionnaires are used in less than half of the studies. Baselines for comparison are included in about one half of the studies, but placebo studies are rare. At least a minimal discussion of the study's limitations is included in about one half of the studies.

### 4.4. Evaluation Type and Fidelity

The results for the categorization of user studies by evaluation type are shown in Figure 5. Most studies rely on subjective satisfaction or subjective comparison of methods. However, the top 4 group user studies feature more forward simulatability and less subjective satisfaction. The results are similar to the results in Nauta et al. (2023) and the discrepancies can be explained by the difference between the two samples of papers and by our interpretation of subjective comparison (see Appendix A.2.1 for details). The results also align with the results of our additional categorization into objective or subjective evaluation (see Figure 6). While subjective evaluation is more common, both types are common and the difference is less in the top 4 group.

The fidelity and evaluation level results of our additional categorization are shown in Figure 7. Most studies evaluate the user's understanding of the AI system and do so on a toy application or absent an application context. The third

most common category are medium fidelity user studies that evaluate downstream performance. There are very few high fidelity user studies. There are no discernible differences between user studies up to 2020 and 2021 and after. However, results suggest that user studies from the top 4 group are more frequently conducted without an application context and focus on understanding.

## 5. Alternative Views

Our view, that the state of user studies in XAI is relatively poor, is generally accepted and we provide further empirical evidence. Similarly, we do not believe that it is controversial to state that the field of ML should hold itself to high methodological standards when it comes to user studies (see Herrmann et al. (2024) for a similar sentiment regarding empirical research in ML in general). However, there is an alternative view that user studies are not as key for the development of XAI as we claim.

Arguments against user studies are always based on a comparison with the only alternative, which is evaluating XAI without users (or functional evaluation). The two major points of criticism are (a) that user studies are more difficult to do and (b) that they are biased.

It is undisputed that user studies are more difficult to do than functional evaluation. However, if a user study is appropriate for the task at hand and functional evaluation is at most a compromise (and potentially inappropriate), the difficulty of conducting a user study is by itself not a valid argument for using functional evaluation.

When considering this argument and other arguments that

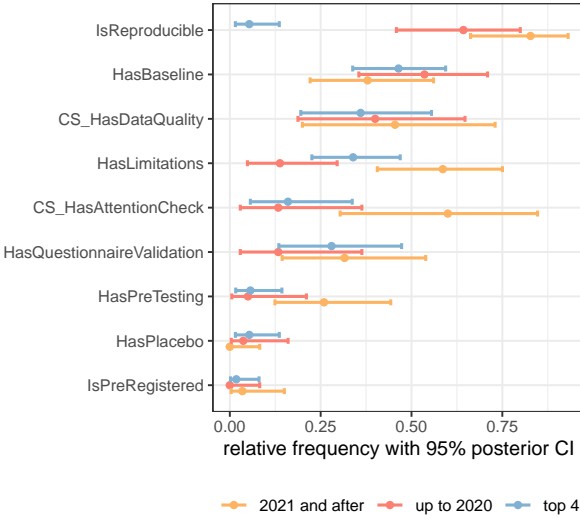

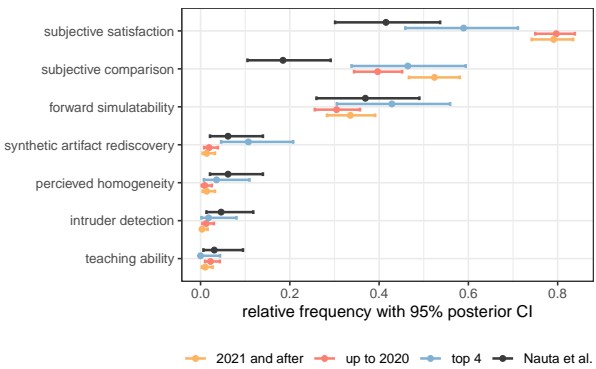

*Figure 5.* Relative frequencies of quantitative evaluation type categories. We include the results from (Nauta et al., 2023), who introduced the categories. Note that their sample were user studies from 12 top CS/ML/AI venues in the period from 2014 to 2020.

*Figure 4.* Relative frequencies user study properties. Note that these are based on random subsamples (with replacement) of size 30 for the up to 2020 and 2021 and after groups.

follow in this section, we should also take into account that there is little to no empirical evidence that validates functional evaluation metrics via downstream task performance. In fact, there is evidence to the contrary and evidence that even in user studies the performance of XAI depends strongly on context. Therefore, proxy tasks or any other deviation from the real-world context can result in misleading results (see Section 3.3 for details).

The other major point of criticism against user studies is bias (Alangari et al., 2023; Kadir et al., 2023). This is most commonly expressed as follows: functional evaluation is objective, while user studies are subjective. With the implied understanding that objective is better than subjective.

For example, Petsiuk et al. (2018) argue that keeping humans out of the evaluation makes it more fair and true to the classifier's own view of the problem, rather than representing a human's view. Rong et al. (2023) state that functional and human-subject based evaluation address two different things. One addresses the general objectivity independent of downstream tasks, while the other contextualize with specific use cases. Markus et al. (2021) write that although quantitative proxy metrics are necessary for an objective assessment of explanation quality, they should be complemented with human evaluation methods before employing AI systems in real-life. Due to bias, Kadir et al. (2023) call for a functional evaluation metric that can be experimentally validated.

Alangari et al. (2023) also argue that, as a consequence of the limitations of user studies, there has been a decline in

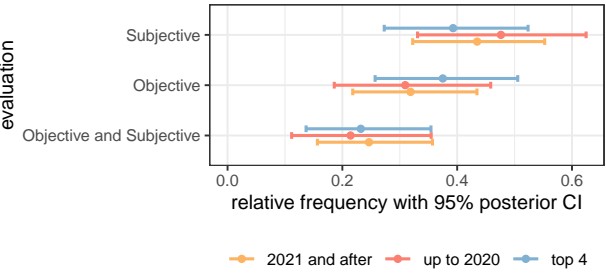

*Figure 6.* Relative frequencies of objective and subjective evaluations and 95% posterior CI. Note that these are based on random subsamples (with replacement) of size 30 for the up to 2020 and 2021 and after groups.

the use of user studies, with functional evaluation gaining prominence as a more rigorous approach. According to Nauta et al. (2023) the proportion of user studies in all evaluation has remained relatively steady from 2016 to 2020 (around 20%). Our data would support the interpretation that the number of user studies has tapered off since, while the number of papers in XAI keeps growing. However, even if that is the case, we would rather attribute this to the fact that functional evaluation is easier to conduct and not to the limitations of user studies.

We argue that the bias inherent to user studies is by itself not a strong argument against user studies or in favor of functional evaluation. It overlooks the fact that functional evaluation also introduces a bias when used as a proxy for some downstream performance. If we agree that the end goal is to satisfy societal desiderata, any functional evaluation should be validated via user studies (and thus

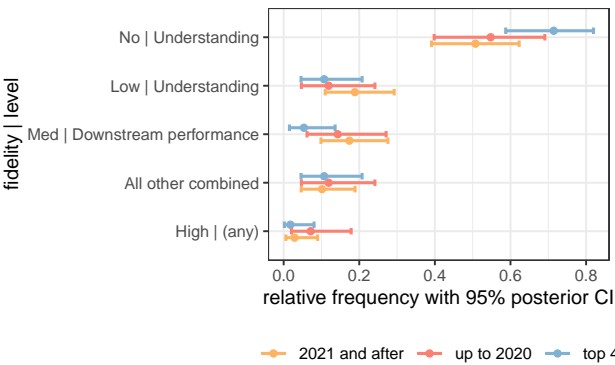

*Figure 7.* Relative frequencies of the most common combinations of fidelity and level and 95% posterior CI. Note that these are based on random subsamples (with replacement) of size 30 for the up to 2020 and 2021 and after groups.

subjectively). So, user studies, subjective or not, cannot be avoided in a field that is supposed to be focused on humans.

Functional evaluation also has a major limitation that has so far been overlooked. A functional evaluation method cannot be used to compare XAI with different types of outputs. For example, to compare feature importance with counterfactuals or a heatmap with a text-based explanation. Nauta et al. (2023, Table 2) categorize almost 100 example methods, all of which are tailored to specific outputs and cannot be used to compare different types of explanation methods. The underlying issue is that the output (that is, the presentation) of the XAI method is a fundamental part of the method itself, so we cannot standardize outputs without changing the method. Unlike, for example, the task of classification, where prediction outputs (class or probabilities) put all models on the same denominator and simplify comparison. The same issue makes it more difficult to derive placebic explanations.

As an alternative view, we could also adopt the position that XAI is still progressing, thereby circumventing the discussion about potential issues with functional evaluation and the necessity of user studies altogether. That is, if progress continues despite these issues, quality user studies cannot be essential to that progress.

If we measure progress by how much XAI methods are being used, XAI has definitely been progressing. We acknowledge that XAI has produced useful tools for ML practitioners, but would add that in the case of the ML practitioners, the researchers and developers are often the target users, so it can be argued that these methods are being developed close to the target user.

The use of XAI methods has been growing outside of ML as

well, due to the popularity of feature contribution methods, in particular SHAP. However, it is not clear if this increase in popularity is due to the progress of XAI or due to the increase in the use of AI in general. And, unlike tabular predictive modeling, natural language processing, or computer vision, where recent developments are already being used in practice, that is not the case with XAI.

Even if we take the position that increased use of XAI methods implies that these methods are useful for users other than ML practitioners, it is not clear how we can reconcile this with growing evidence that the tools being developed are not what target users need (see Section 2.1). And even then we at a minimum have to acknowledge that we have little to no understanding of why, when, and for which type of target user they are useful.

Finally, note that our position does not reject functional evaluation. On the contrary, functional evaluation should play an important role in verifying whether a method meets minimal technical criteria, particularly during rapid prototyping and the early phases of development Miller et al. (2017). It could also serve as a less resource-intensive alternative to user studies, provided that we establish clear links between functional evaluation and the satisfaction of desiderata or performance on downstream tasks.

## 6. Conclusion

We share the view of Herrmann et al. (2024), who, in their position on empirical research in ML, call for more confirmatory research, comparison studies, replication studies, and meta-studies. And, that the field should *move from being largely driven by mathematical proofs and application improvements to also becoming a full-fledged empirical field driven by multiple types of experimental research.*

In this paper we focused on XAI and on user studies - one type of empirical research that we believe to be key for the advancement of XAI. Historically, the field of ML has relied on theoretical work and improvements over the state-of-the-art with respect to some abstract metric. And in most cases that has led the field very far. However, unlike, for example, supervised learning, where predictive performance is easily measured and plausibly translates to real-world utility, the same cannot be said for XAI. In XAI, the user is an integral component that cannot be easily circumvented, and any attempt to do so risks widening the gap between academic research and practical application.

User studies in XAI are poorly designed (as a whole, with some exceptions) and have (with a few exceptions, such as Evirgen & Chen (2022); Kiani et al. (2020); Kenny et al. (2024); Wong et al. (2024)) low or no fidelity to real-world use. As a result, at best, we know how a ML practitioner's understanding of a model improves with XAI (through self-

reporting and forward simulatability). At worst, we know very little about the practical application and real-world benefits of XAI.

The design and fidelity problem appears to be exacerbated in top ML venues, where it is very difficult to publish a paper that focuses solely on a user study, but at the same time more poorly-designed user studies get published as a part of theoretical or methodological papers. This at least exhibits a consistent view that user studies are not considered that important at these venues. However, we believe that user studies that clearly do not meet even the minimum methodological standards of reproducibility should be rejected, regardless of the focus of the venue. We also believe that methodological developments in XAI that rely exclusively on functional evaluation (as opposed to user studies or theoretical justification) should be subject to greater scrutiny, to deal with the field's overreliance on functional evaluation.

Our results also suggest that the level of relevant knowledge and know-how in user studies is relatively low in the ML community, not only in conducting but also in reviewing them. This is understandable, given the field's history and focus. Some may even argue that other fields, such as psychology and HCI, should take the lead in researching XAI in practice. In particular, researchers from those fields. However, we believe this would be a missed opportunity to elevate the quality of research within the ML community.

How can we encourage better user studies? Recently, venues have emerged to promote empirical research, such as the Journal of Data-centric Machine Learning Research (DMLR), the Datasets and Benchmarks Track at NeurIPS, and the Applied Data Science Track at ECML. A similar venue dedicated specifically to user-centric research in ML would be invaluable. Such a platform could not only foster user-centric research but also enhance the community's expertise in conducting and evaluating user studies. This, in turn, would improve the quality of reviews and elevate the standards of published user studies.

We conclude with a list of XAI topics that we believe are particularly important for the advancement of the field:

- Linking functional evaluation metrics, proxy tasks, and real-world performance. In particular, the development of standardized benchmarks.

- Generalizability of crowdsourced and student-based studies.

- Intra-user and inter-user differences.

- The placebo effect and the development of placebo-controlled studies.

- Neutral method comparison studies of popular XAI methods. We anticipate that the results in the field are

overly optimistic, so we share the sentiment of Karl et al. (2024) that we should embrace negative results.

## Acknowledgements

We sincerely thank the anonymous ICML reviewers for their insightful feedback, which helped us clarify our contributions and significantly improve the paper. This work was funded by the Slovenian Research Agency (research core funding No. P2-0442).

## Impact Statement

This paper presents a position on the importance of user studies in the evaluation of explainable AI. Its goal is to spark discussion and raise the standards of user-centric research in explainable AI and in machine learning in general. There are other potential societal consequences of our work, none of which we feel must be specifically highlighted here.

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

# A. Empirical Analysis of XAI Papers with a User Study

Here we provide details on how we collected and analyzed the XAI papers that were the basis for the empirical analysis in Section 4. The list of all 607 papers with meta-data is available for download[1].

## A.1. Literature Search

We performed three different searches, with Scopus as the starting point in all three. The inclusion criteria were that the paper has a user study (no restrictions on the type or number of participants) and is from the broader field of XAI (explaining AI or ML systems or models; no restrictions on the type of explanation).

### A.1.1. DIRECT SEARCH

We performed a search through Scopus on Apr 7, 2024 with the search string:

```
TITLE-ABS-KEY(("explainable AI" OR "XAI" OR "interpretable AI" OR "interpretable
    machine learning")) AND (participant* OR "user study" OR "user evaluation*" OR "
    user rating*" OR "subjective rating*" OR "human evaluation*" OR "human study" OR
     "human rating*")
```

The query returned 865 candidate papers. We manually checked these papers and found 221 that met the inclusion criteria.

### A.1.2. INDIRECT SEARCH

Next, we searched for XAI papers with a user study indirectly, through XAI papers that were a survey of evaluation in XAI or contained, as part of related work, a collection of XAI papers with a user study.

We performed a search through Scopus on Apr 7, 2024 with the search string:

```
TITLE-ABS-KEY((("explainable AI" OR "XAI" OR "interpretable machine learning")) AND
    ("survey" OR "review") AND "evaluation")
```

The motivation for this search was threefold. First, it doubles as a search for the most related work. Second, it ensures that our collection of papers includes at least all of the papers that are referenced in the most related work. And third, some XAI papers with a user study are difficult to find with a keyword-based search, because they do not contain any of the typical keywords.

The search query returned 192 candidate review papers, 7 of which had a literature review with at least some focus on user studies. We performed forward and backward snowballing from these 7 papers, until we found no new review papers. This resulted in 18 papers: Alangari et al. (2023), Bertrand et al. (2023), Chromik & Schuessler (2020), Ferreira & Monteiro (2020), Herm et al. (2022), Johs et al. (2022), Keane et al. (2021), Lai et al. (2021), Liao & Varshney (2021), Lopes et al. (2022), Mohseni et al. (2021), Nauta et al. (2023), Nguyen et al. (2024), Qian et al. (2023), Rong et al. (2023), Sperrle et al. (2021), Williams (2021), and Zhou et al. (2021). Note that the three papers with the most user studies referenced are Herm et al. (2022) (152, only 25 referenced in the paper), Rong et al. (2023) (97), and Nauta et al. (2023) (65).

We manually checked the papers referenced in the above 18 papers and found 358 that met the inclusion criteria and were not found with direct search.

### A.1.3. TOP CONFERENCES SEARCH

Finally, we searched for XAI papers with a user study in four of the top venues for ML research (AAAI, ICLR, ICML, and NeurIPS) from 2020 to 2024.

We performed a search through Scopus on Oct 26, 2024 with the search string:

---

[1]https://github.com/estrumbelj/XAI-user-studies-dataset/blob/main/dataset.csv

```
TITLE-ABS-KEY ( ( "explain*" OR "XAI" OR "interpret*" OR "explan*" ) ) AND ( PUBYEAR
    > 2019 ) AND SRCTITLE ( "ICML" OR "Advances In Neural Information Processing
    Systems" OR "AAAI" OR "ICLR" ) AND ( questionnaire OR crowdsourc* OR "amazon
    mechanical turk" OR "prolific" OR participant* OR "user stud*" OR "user eval*"
    OR "human subject*" OR "user rating*" OR "subjective rating*" OR "human eval*"
    OR "human stud*" OR "human-subject*" OR "subjects" OR "human rating*" )
```

The motivation for this search was twofold. First, these venues are of particular interest, because of their impact on the ML community, both in terms of research directions and standards. And second, a narrower scope of venues allowed us to relax the search string and obtain a larger and more systematic subsample for this subset of venues.

The query returned 359 candidate papers. Note that the query returns papers from conferences other than the four target conferences. We did not include such papers. Also note that NeurIPS 2024 was not indexed by Scopus at the time of the search.

We manually checked the papers and found 47 that met the inclusion criteria. Out of these 47 papers, 28 were newly found and 19 were already found in the previous two searches. Note that nine papers from these four venues were found in the indirect search but not in this search. As expected, all papers found in the direct search were also found in this search, because the direct search had a strictly more restrictive search query.

### A.2. Analyzing the Papers

The additional data on the XAI papers with a user study are summarized in Table 1. To make the workload manageable, some of the data that require manual review are included only for a subsample of 116 papers. We sampled (with replacement) 30 papers published up to 2020, 30 papers published 2021 or later, and all 56 papers from the four conferences and published 2020 or later. We used a simple Binomial-Beta Bayesian model with Beta($\frac{1}{2}, \frac{1}{2}$) to infer the proportions and we report 95% posterior CI based on 2.5% and 97.5% quantiles.

#### A.2.1. CATEGORIZING THE TYPE OF QUANTITATIVE EVALUATION

To categorize the type of qualitative evaluation methods used in XAI user studies, we used the 7 evaluation methods identified by Nauta et al. (2023, Table 5).

For *forward simulatability*, *intruder detection*, and *perceived homogeneity* we were able to follow the original definitions.

For *teaching ability* we expanded the original definition, where the participant should be able to predict new instances without explanations, with cases where participants learned a valuable skill (for example, children learned how to better handle their diabetes).

For *synthetic artifact rediscovery* we only included a study if the artifact was added later. For example, if the model was changed to include gender bias, which was not originally there. For example, if the user was to perceive whether or not the model was biased, we counted this as subjective satisfaction.

For *subjective comparison* we included only user studies that compare two or more XAI methods. We included even small changes in presentation. For example, showing feature importance with or without overall model accuracy. We did not include studies that compared different ML models but all with the same type of explanation.

For *subjective satisfaction* we added two groups of user studies. The first are XAI frameworks which allow users to do exploratory analysis, after which the users describe how the framework helped them. The second are user studies that interview participants for their opinion, without an actual application of XAI. Note that fairness is a commonly measured subjective satisfaction item not listed in the original definition.

Our application of the categories resulted in 1.8 categories per paper, compared to 1.2 categories per paper in Nauta et al. (2023). This can be partially explained by our more liberal interpretation of subjective comparison. Furthermore, Nauta et al. (2023) focused on top ML venues, where our results are very similar. That is, the discrepancy can be explained by more frequent use of subjective comparison and subjective satisfaction in venues outside of top ML venues.

*Table 1.* A summary of the data for the XAI papers with a user study.

| | COLUMN | NOTES |
|---|---|---|
| PAPER INFO | TITLE
DOI
VENUE
ABSTRACT
PUBLICATIONYEAR | NA if not available. |
| SAMPLING INFO | WHICHSEARCH
SUBSAMPLINGGROUP | Which search found this paper (direct, indirect, top 4).
NA if the paper was not subsampled for analysis, otherwise its group (up to 2022, 2021 and after, top 4). Comma-separated if more than one applies. See A.2. |
| USER STUDY (ALL) | PARTICIPANTTYPE | User study participant type (administrative staff, AI expert, crowdsourcing, domain expert, researchers, students, other, not stated). Subtype provided in parentheses (for example, which crowdsourcing platform). Comma-separated if more than one type. |
| | PARTICIPANTCOUNT | NA if not available. Comma-separated if more than one user study. |
| | NAUTACLASSIFICATION | Comma-separated if more than one type. See A.2.1. |
| USER STUDY (SUBSAMPLE) | CATFIDELITY
CATLEVEL
CATOBJSUBJ
HASBASELINE
HASPLACEBO
ISPREREGISTERED
ISREPRODUCIBLE | See A.2.2.
See A.2.2.
See A.2.2.
See 3.4.
See 3.4.
Was the study pre-registered.
Is the study reproducible. True, unless it is missing key information. |
| | HASPRETESTING | Does the study report any pretesting or pilot study before the main study. |
| | HASQUESTIONNAIREVALIDATION | Are the user measurement instruments validated in this or a previous study. |
| | HASLIMITATIONS | Does the study have at least a minimal discussion of limitations. |
| | CS_HASATTENTIONCHECK | Crowdsourcing user studies only. Does the study include any type of attention check. |
| | CS_HASDATAQUALITY | Crowdsourcing user studies only. Does the study include any type of post-collection data quality assurance. |

### A.2.2. CATEGORIZING USER STUDIES ON FIDELITY

User studies vary in how well their findings translate to real-world applications. Therefore, when evaluating the real-world performance of XAI, it is helpful to score or categorize user studies on this dimension. We introduce four categories for the *fidelity* of the user study setting to a real-world application:

- *High:* XAI is embedded in a real-world application and evaluated in a real-world setting. Example: Millecamp et al. (2019) tested their XAI system by directly connecting to the participant's Spotify account to provide them with song recommendations and corresponding explanations.

- *Medium:* Suggests a clear real-world application, but it is not evaluated as such. Example: Paleja et al. (2020) propose a framework for scheduling. The authors do provide an example of a real-world use (Taxi domain), which demonstrates a plausible use for use in real-world situations.

- *Low:* Embedded in a toy or mock application. Example: Ben David et al. (2021) aim to develop a financial algorithmic advisor and evaluate their approach on a simplified lemonade stand game.

- *No:* Not embedded in an application. Example: Taesiri et al. (2022) tested their explanation on a dataset, but there is no sign of how the explanation could be used in real life or what motivates the choice of dataset.

We complement fidelity with the evaluation *levels* inspired by the model by Speith & Langer (2023):

- *Explanatory:* Evaluating how accurately XAI describes the AI system. While a user study can be used for this level, we did not find any in the subsample. That is, functional evaluation is typically used for this level.

- *Understanding:* Evaluating how well the XAI facilitates understanding of the AI system. This can be measured *objectively* (for example, by forward simulation) or *subjectively* (for example, self-reported understanding or trust).

- *Downstream performance:* Evaluating how well XAI contributes to the task the AI system is designed to solve. Can be measured objectively (for example, task performance) or subjectively (self-reported confidence).

Note that we also considered using two existing categorizations: the most popular taxonomy for XAI evaluation methods by Doshi-Velez & Kim (2017; 2018) and the Technology Readiness Levels (TRL) scale, a well-known scale for estimating the maturity of technological development developed by NASA. The taxonomy by Doshi-Velez and Kim subdivides uses studies into application-grounded (real humans, real tasks) and human-grounded (real humans, simple tasks). We found this to be insufficiently granular for our purpose. The TRL scale or its adaptations to AI (Browne et al., 2024; Lavin et al., 2022) or human readiness level (HRL) (See, 2021) are more granular (typically a 9-point scale). However, their purpose is to estimate technology readiness during system development by qualified experts on the design team (Handley et al., 2024). Post-hoc application of this scale to user studies is difficult, because there is in most cases no clear application context, incremental progress, or end goal.

### A.3. Limitations

Systematically searching for XAI papers with a user study is difficult. XAI papers might not contain any of our XAI or interpretable ML search terms. Furthermore, they might not even contain any of our user study or participant search terms. The limitations of keyword search are clear from the large number of papers found by indirect search but not direct search. A quick manual inspection of 20 randomly chosen papers found by indirect search showed that 19 of them were indexed by Scopus. Therefore, most papers not found by direct search were not found because they did not contain the search terms (assuming that there are no issues with Scopus data or search).

The chosen search terms were a tradeoff to limit the manual inspection of papers for inclusion to the order of 1000s. The only better alternative would be to manually inspect all papers, which we do not find feasible, unless we severely limit the number of venues and the time period.

As a result, our sample of papers is biased in several ways. It inherits any biases towards venues and publication years of Scopus. However, we argue that Scopus indexes the vast majority of relevant venues, especially in the period of the past 5 years. Indirect search references can only go back in time, so the sample of papers found by indirect search is biased towards older papers. We partially mitigate this by splitting the papers into two groups based on publication year. The indirect sample is also biased towards certain venues. The data we collected does allow us to partition the papers by venue, but we have not done so. Similarly, we could limit our analysis to direct search papers only, which would mitigate at least the biases of indirect search.

In the context of our position, the key question is whether these biases also result in a bias towards lower quality studies. We believe it is very unlikely that a paper that is more easily identified as a XAI paper with a user study, is more likely to have a poorly designed user study. However, it is likely that newer papers have better user studies, which we do investigate by splitting the papers into two groups with respect to time.

Some of the user study variables were not trivial to measure (NautaClassification, IsReproducible, ParticipantType, CatFidelity) and, while we do believe the data are measured relatively consistently (a single rater for each variable), it is possible that there is some between-rater variability. At least for the key claims, this should be mitigated by the fact that when in doubt, we chose to benefit the quality of the user study.

To summarize, this is an exploratory study. A more systematic confirmatory study is required to further validate the findings of this exploratory study. For a more complete view of the quality of user studies in XAI, we also lack a more diverse set of venues, in particular, human-centric research venues (for example, FaaCT) and other top AI/ML venues (for example, IJCAI). However, we argue that the sample is still large enough to draw conclusions and to support our position. For example, if most of the found user studies at top venues are not reproducible, this is reason for concern. Even if we allow for the unlikely scenario that all user studies that we did not find are reproducible.

