# OpenReview forum: "Position: Explainable AI Cannot Advance Without Better User Studies"
_ICML.cc/2025/Position_Paper_Track — ICML 2025 Position Paper Track poster_

### Official Review · Reviewer_1mob · 2025-02-19

**Significance:** 4
**Argument Clarity:** 3
**Rating:** 4
**Confidence:** 5

**Questions:**

No questions aside from above

**Discussion Potential:**

4

**Paper Summary:**

The paper does a large survey of user studies in the XAI literature. It particularly focuses on the "Top 4" venues of NeurIPS, ICML, ICLR, and AAAI. Several points are made which mostly centre around the idea that we should be embracing user studies as the future in XAI research, more so than more technical methods. Some suggestions are finally made for researchers moving forwards at the end such as placebo controlled studies etc.

# Summary after rebuttal
Nothing to add here past the rebuttal discussion, I encourage the authors to e.g. define placebo-controlled better, but otherwise nothing to add, all the best.

**Position:**

Yes

**Position In Title:**

Yes

**Related Work:**

2

**Strengths And Weaknesses:**

### Strengths
* The paper has some novel ideas I hadn't really considered before such as trying to link functionally grounded explanations with application focused utility.
* I also like the point about how MTurk workers (or Prolific) are generally biased also towards certain demographics, something I had considered myself but the community doesn't really acknowledge.
* A few more important points are brought to the fore, such as the difficulty of publishing pure user centered research at these "Top 4" venues.

### Weaknesses
* I know the focus is on ML conferences, but I'm not really sure I agree with the authors notion of Top 4 venues, you really need to better justify why these are "the best", especially when we're considering user centred research (they are generally not considered the best here).
* Some figures are bad, Figure 1 for example has no legend describing what the blue highlights are, take better care with this please.
* I think the paper would benefit from a section going over successfully conducted application focused research which has shown good results for XAI, for example take these two papers [1, 2], these studies are rare in that they show practical real world usefulness of XAI, and would give the paper (and readers) some "hope" for the future of the field.
* I don't really understand what you mean by "placebo-controlled" studies? This should be better explained, for example this study [3] shows two explanation options to people, one of which is a "bad explanation placebo" on purpose and measures accuracy as you discuss, is that what you mean?
* Line 221: Qualtrics is not a crowdsourcing platform, it is a survey maker, these two things should be properly distinguished.
* There are also some citations I think would be useful, Line 97 [4] also make a passionate call for more user studies, Line 126 [5,6] also heavily discuss mental models and how they can be split up and vary over time in a study, might be nice to talk about here so we can all build up together as a field.

Apart from all that, I think the biggest weakness is that the ML community in general has "passed the buck" to the HCI community etc. to do these studies, so it could cynically argued there's no need for this paper. However, I do agree this is bad practice, and we should instead be encouraging this kind of research, at least in an interdisciplinary way. If the authors address my concerns in the revision, I will raise my score.

***
[1] Kenny, E., Dharmavaram, A., Lee, S., Phan-Minh, T., Rajesh, S., Hu, Y., Major, L., Tomov, M. and Shah, J., Explainable deep learning improves human mental models of self-driving cars.

[2] Wong, F., Zheng, E.J., Valeri, J.A., Donghia, N.M., Anahtar, M.N., Omori, S., Li, A., Cubillos-Ruiz, A., Krishnan, A., Jin, W. and Manson, A.L., 2024. Discovery of a structural class of antibiotics with explainable deep learning. Nature, 626(7997), pp.177-185.

[3] Kenny, E.M., Tucker, M. and Shah, J., 2023. Towards interpretable deep reinforcement learning with human-friendly prototypes. In The Eleventh International Conference on Learning Representations.

[4] Keane, M.T., Kenny, E.M., Delaney, E. and Smyth, B., If Only We Had Better Counterfactual Explanations: Five Key Deficits to Rectify in the Evaluation of Counterfactual XAI Techniques.

[5] Kenny, E.M., Ford, C., Quinn, M. and Keane, M.T., 2021. Explaining black-box classifiers using post-hoc explanations-by-example: The effect of explanations and error-rates in XAI user studies. Artificial Intelligence, 294, p.103459.

[6] Hoffman, R.R., Mueller, S.T., Klein, G. and Litman, J., 2018. Metrics for explainable AI: Challenges and prospects. arXiv preprint arXiv:1812.04608.

**Support:**

4

---

> ### Author Rebuttal · Authors · 2025-03-29
>
> We thank the reviewer for the comments, they’ve been most helpful in improving the paper!
>
> **I know the focus is on ML conferences, but I'm not really sure I agree with the authors notion of Top 4 venues…**
>
> We were not aiming for top user-centric venues but top AI/ML venues, in order to demonstrate the lower standards for user studies and fewer user studies (and thus overreliance on functional evaluation). But we agree that there are other candidates for top 4 AI/ML and that user-centric venues would also be of interest (see response to reviewer tsJw for more details).
>
> **Some figures are bad, Figure 1 for example…**
>
> We added a key to the figure. We also double-checked that all other figures have appropriate keys, axis labels, consistent use of color, and captions.
>
> **I think the paper would benefit from a section going over successfully conducted application focused research which has shown good results for XAI, for example take these two papers [1, 2], these studies are rare in that they show practical real world usefulness of XAI, and would give the paper (and readers) some "hope" for the future of the field.**
>
> We added a paragraph listing these and some other high-quality / high-impact user studies from our dataset, highlighting them, as the reviewer also points out, rare exceptions to the rule.
>
> **I don't really understand what you mean by "placebo-controlled" studies?....**
>
> For the most part, we can think of this exactly as the placebo problem in medicine. There is legitimate (to some extent empirically demonstrated) concern that merely administering XAI will improve performance, even if the XAI does nothing. This is a big concern when measuring self-reported trust, confidence, etc., less so if measuring downstream task performance. Therefore, it makes sense to have a placebo-controlled study, where we have a treatment group that receives the actual XAI and a placebo group that receives something that resembles the XAI method in the way it is administered, but it holds little to no information.
>
> So, the problem is definitely real and consequential. Where things get a bit murky is what does it mean to administer something that resembles the XAI, but it holds little to no information. That is, what is a placebo explanation in the context of XAI? In medicine it is easy to create a placebo pill or injection that the participant can’t distinguish from the true treatment. In XAI researchers are only beginning to explore this problem, so the best we can do is list a few examples.
>
> Regarding reference [3]: The “non-human friendly” group’s random rearrangement of prototype images could be considered an attempt at a placebic explanation (looks the same, but holds little to no information). It is akin to the technique of perturbing feature contributions, which leads to the same presentation but mostly nonsensical information. We included [3] in the paper as another example of placebic explanations. The “no explanation” explanation is, as the name already suggests, a no-explanation baseline, which is much more common.
>
> **Line 221: Qualtrics is not a crowdsourcing platform…**
>
> We added the distinction “..crowdsourcing platforms and survey platforms that offer online sample and panel services…”... and then later “CloudResearch (Connect), and Qualtrics (Qualtrics Panels)”.
>
> **There are also some citations I think would be useful…**
>
> We’ve added these three references as suggested (one of them was already cited, but in another context).
>
> **Apart from all that, I think the biggest weakness is that the ML community in general has "passed the buck" to the HCI community etc.**
>
> We would argue that, yes, we’ve passed on the inconvenience of user studies, but not the convenience of churning out new XAI methodology, supported only by functional evaluation or anecdotal evidence (see response to reviewer **tsJw** for more details).

---

### Official Review · Reviewer_tsJw · 2025-02-23

**Significance:** 4
**Argument Clarity:** 4
**Rating:** 4
**Confidence:** 4

**Questions:**

Literature review in section 4: Why exclude conferences such as IJCAI and FAccT?

**Discussion Potential:**

2

**Paper Summary:**

The paper argues that user studies are indispensable for advancing XAI.

**Position:**

Yes

**Position In Title:**

Yes

**Related Work:**

3

**Strengths And Weaknesses:**

## General comments
The fact that there seems a recent significant increase in the number of user studies in XAI, looks to me like the XAI community is aware of the necessity of user studies. Of course, the quality of such studies may vary -- also a lack of education on how to do user studies property. It seems to me that there is no need to tell people that user studies are relevant but there seems definitely a need for training (such as tutorials) people on how to do user studies. I assume that the authors are familiar with the design of user studies, I therefore encourage them to go to those AI/ML conferences and give tutorials on how to properly do user studies. I think many people with a background in AI or ML simply lack the knowledge of how to do a user study.

## Strengths
A timely and important issue is discussed. The paper is easy to read and the authors provide extensive evidence for their position and clearly state their arguments the significance of user studies in XAI.

## Weakness
Not sure how relevant this position is for ICML. ICML is a more mathematical-focused conference -- I am a bit concerned that significant parts of the audience would not value this contribution. Might be a better fit for FAccT or IJCAI. Yet, people at ICML would be definitely an important target group of this position paper -- I would give it a try and see how it is perceived.

**Support:**

4

---

> ### Author Rebuttal · Authors · 2025-03-29
>
> The reviewer’s main concern seems to be that our position that “XAI can’t progress without better user studies” might not be the best fit for the more mathematical ICML conference (and community). And, as such, the paper might lack discussion potential.
>
> We appreciate this concern, because it helped us to rethink and communicate our position more clearly now. Other reviewers also raised similar concerns, so we’ll try to address them all here.
>
> We strongly believe that the warning in our position is not only a good fit for the more mathematical parts of the AI/ML community, but that they are the key target audience, because they (we) are part of the problem.
>
> As reviewer **1mob** points out, for XAI user studies, we seem to have "passed the buck" to the HCI community (or someone else down the line). However, we did not pass all of XAI research or even all the empirical XAI research. Methodology papers supported only by anecdotal evidence or functional evaluation still feature prominently, outnumbering user studies by far.
>
> It seems that the AI/ML community wants to have their cake and eat it too. We eliminate the user from explainable AI research, because user studies are inconvenient, but we still produce XAI methods. However, the gap between academia and practice is now clear, as is the fact that there is no established link between functional evaluation and practical performance.
>
> How exactly then are all the functional evaluation papers contributing to the field of XAI? It is our position that not that much and that publishing so many might even be counter-productive. As we already point out in the paper, XAI is not like other standard AI/ML fields, such as predictive modeling, where functional evaluation (for example, predictive accuracy) has direct implications for practical performance.
>
> There is also a certain amount of prestige associated with top AI/ML venues. Papers published there serve as a reference to many researchers. The messages that are currently being sent are that (a) there is little value assigned to user studies, (b) there is no standard for user studies, and (c) the value of a XAI method is determined by the bolded line in the functional evaluation table. We believe that rejecting (or removing) substandard user studies and appreciating quality user studies, especially those that justify functional evaluation, would send a message that is more likely to move the field of XAI forward.
>
> Finally, we don’t believe that lower-quality user studies are due to a lack of resources on how to conduct a high quality study.
> The reviewer also comments that a recent significant increase in the number of user studies in XAI might indicate that the field is becoming more aware about the importance of user studies. We are not sure that that is the case. While there is strong evidence that the number of user studies is growing, there is no evidence that the quality of user studies is improving or even that the growth is at least proportional to the overall growth of the field of AI/XAI. That is, we think it is more plausible that we have more user studies, because we have more people working in the field.
>
> We incorporated the above points into the paper, to make our position more clear.
>
> **Q1: Literature review in section 4: Why exclude conferences such as IJCAI and FAccT?**
>
> Several papers from these two venues were collected in the 1st part of our literature search and are included in our dataset (and in the non-top 4 groups in the analysis). In principle, anyone can reproduce the results for the subset of venues of their interest, for example, more user-centric venues such as FAccT. The main reason why we only did a separate analysis for the top 4 group is that the papers for that group were collected in a more systematic way.
>
> Why didn’t we pick IJCAI and FAccT in our top 4? Our focus was on “top” AI/ML conferences and we believe that NeurIPS, ICML, ICLR, and AAAI are a reasonable choice. We agree that an argument could be made for several other venues. If we covered more, we would have included at least IJCAI. However, it takes a lot of effort to thoroughly search even 4 years for one of these venues, so we had to draw the line somewhere.

---

> > ### Comment · Reviewer_tsJw · 2025-04-02
> >
> > Thanks for the clarification.
> >
> > I understand that you selected NeurIPS, ICML, and ICLR because ICML reviewers are most likely from this community. Still, I think if you include AAAI, you must also include IJCAI -- AAAI is not "better" or "more prestigious" than IJCAI. I understand that including every conference is not feasible.
> >
> > I still like the paper and think it can have an impact on the community. I will raise my score to increase your chances of acceptance.

---

> > > ### Author Response · Authors · 2025-04-02
> > >
> > > Thank you for reading our rebuttal and supporting our paper!
> > >
> > > We agree with the criticism of our choice of top 4 venues. Unfortunately, we can't just add IJCAI papers to the top 4 results, even the ones we have in our sample, because they were not sampled the same way as the top 4 group papers and not all were categorized (while this can be done, it can't be done in a short period of time).
> > >
> > > We added to the limitations part of the study that: *A more systematic confirmatory study is required to further validate the findings of this exploratory study. For a more complete view of the quality of user studies in XAI, we also lack a more diverse set of venues, in particular, human-centric research venues and other top AI/ML venues (for example, IJCAI).*
> > >
> > > We are actually already planning such a study.

---

### Official Review · Reviewer_SwGW · 2025-02-25

**Significance:** 2
**Argument Clarity:** 3
**Rating:** 2
**Confidence:** 4

**Questions:**

All clear.

**Discussion Potential:**

2

**Paper Summary:**

The paper essentially argues that XAI is currently weak in user studies, and supports this with a quantitative study of published works.

**Position:**

Yes

**Position In Title:**

Yes

**Related Work:**

3

**Strengths And Weaknesses:**

I think the paper has two key weaknesses:

1. the point that user studies are a weakness of XAI research is hardly new, which is also exemplified by the last paragraph of section 2 that lists many papers that also arrived at that conclusion. The authors themselves write "Our view that the state of user studies in XAI is relatively poor, is generally accepted...". They then proceed to counter the point that " functional evaluation is
objective, while user studies are subjective", which they identify as the key argument that could be brought forward against user studies. They essentially claim that functional evaluations also introduce a bias when used as a proxy for some downstream performance, and that the functional evaluations cannot compare XAI with different types of outputs. I don't think that these arguments are particularly novel (or "overlooked" as the authors claim). I am also not entirely convinced by them (a) you could do a functional evaluation on the downstream performance, who says that this should be a proxy? and b) there could also be functional evaluations that apply to different types of models, in the same way as, e.g., accuracy works for a wide variety of models.

2. The paper is not very strong on the claimed position, namely that XAI **cannot** progress without better user studies. One may, e.g., argue that it has already progressed quite a bit, without user studies, to the extent that post-hoc XAI tools such as SHAP are readily available and frequently used in practice. I am not disputing that more user studies would certainly be helpful, but the stated position is that work in AI **cannot** progress without them. Why should we have hit a barrier at exactly this point? I can imagine that this can indeed be argued, but the authors do not try to that in very much depth (the above-mentioned discussion where the authors try to counter arguments that claim that user studies are subjective is as close as they get to that point).

A minor point: The authors argue that "Academic interest in XAI is also growing exponentially". I don't think that this follows conclusively from the provided evidence that half of the XAI papers in Scopus are from 2024. (I am certainly not debating that there is a strongly growing interest, I am just taking issues with the loose use of the word "exponential").

**Support:**

2

---

> ### Author Rebuttal · Authors · 2025-03-29
>
> We thank the reviewer for the comments! Although we disagree on some points, the discussion is very valuable to us.
>
> We also put an effort into making our position more clear (see responses to other reviewers, in particular **tsJw**).
>
> And if the reviewer finds any of our arguments below convincing, we would appreciate it if they point them out, so we can condense them and incorporate them into the paper.
>
> **the point that user studies are a weakness of XAI research is hardly new…**
>
> It is not new, but unlike related work that makes this claim, we integrate all work related to user studies and provide systematic empirical evidence on the quality of user studies and highlight particular weak points, including the lower quality of user studies at top AI/ML venues.
>
> **They essentially claim that functional evaluations also introduce a bias when used as a proxy for some downstream performance, and that the functional evaluations cannot compare XAI with different types of outputs. I don't think that these arguments are particularly novel (or "overlooked" as the authors claim)**
>
> The bias of functional evaluations is well established, yes. But it does not appear that the community has also accepted the consequences. Researchers are still heavily relying on functional evaluation and not putting any effort into understanding this bias by measuring it with user studies, so we believe it deserves more attention.
>
> The inability of functional evaluation to compare two methods of different type might be obvious once pointed out, but we are not aware of papers that discusses this (if the reviewer is aware of one, we would appreciate it).
>
> **I am also not entirely convinced by them (a) you could do a functional evaluation on the downstream performance, who says that this should be a proxy?**
>
> Functional evaluation is evaluation without humans. Whatever we do without humans will always be at most a proxy for the performance of the XAI system with a human involved. And, by definition, there is no XAI without a human involved.
>
> **and b) there could also be functional evaluations that apply to different types of models, in the same way as, e.g., accuracy works for a wide variety of models.**
>
> There is a strong argument against the existence of functional evaluation that can compare fundamentally different types of explanations. We already make that argument in the paper, but we’ll reiterate here: Because XAI is to be consumed by a human, the presentation is part of the method (that is, what you show to the user is the method itself). If we want to functionally compare two different presentations, we have to put them on the same denominator. But by doing so, we are changing the presentation. That is, we are comparing two abstractions, not the presentations consumed by the human.
>
> We may allow for the possibility that such functional evaluation will someday be discovered. But the fact is that it currently does not exist. And seeing how important such functional evaluations would be, this is not due to a lack of trying.
>
> **The paper is not very strong on the claimed position, namely that XAI cannot progress without better user studies…**
>
> To summarize our general argument: XAI methods cannot be well understood without quality user studies. Because there are very few such studies, XAI methods are not well understood. And without good understanding, there can’t be progress.
>
> The first potential criticism is that XAI methods can be well understood with functional evaluation only. The second potential criticism is that there are enough quality user studies. We believe we provide enough evidence against both of these.
>
> The third and final potential criticism is that better understanding is not key for the progress of XAI. This appears to be the argument made by the reviewer. SHAP and similar tools being used a lot constitutes progress. With this we couldn’t disagree more.
>
> Yes, XAI methods have some demonstrated utility in the case of the ML practitioner and model diagnostics (we already acknowledge this in the paper). However, if confronted with questions about who is using these methods and to what effect, one would be hard pressed to find answers in existing literature. And outside of ML practitioners, XAI has failed to provide many practically useful methods or insights. This is also evident from more and more related work pointing out that academia is not solving problems that are relevant for practice.
>
> We also have to take into account that SHAP is an approximation to Shapley values for feature importance, which are about 16 years old. An argument could be made that not much new can be done in the field of feature contribution methods that is beyond academic interest. Yet 100s of feature importance methods have been developed since, with at most functional evaluation and to little practical effect. On the other hand, we don’t understand even the most basic questions, like how many features a typical user can handle.

---

> > ### Comment · Reviewer_SwGW · 2025-04-03
> >
> > As a clarification: I am not saying that XAI does not need to do user studies. It certainly does, and I think every researcher in XAI will agree to that. The problem is not that researchers do not acknowledge this or are not aware of this, but the problem is that user studies are, of course, not easily done, and therefore often proxy measures have to be used.
> >
> > What I still question, however, is the statement that XAI *cannot* progress without better user studies. I don't believe this, and I don't think that this paper (or the rebuttal) supports this strong claim. In many fields of science (say, e.g., aviation or transport), substantial progress is made in laboratory conditions, via simulations. You don't need to construct and fly (and potentially crash) a plane or car in order to improve upon the design. For very much the same reasons, I don't see why we should not be able to make progress in XAI in laboratory conditions. Yes, the ultimate test are user studies (and the plane has to fly in the end), but we can also make progress without them (and, in fact, have made considerable progress without them).
> >
> > But I see that my co-reviewers are more easily convinced. I stand by my original recommendation, which is that I lean towards rejection, but don't mind if this is eventually accepted.

---

> > > ### Author Response · Authors · 2025-04-03
> > >
> > > The reviewer maintains the position that (1) AI/ML community is already well aware of most of the issues pointed out in the paper and (2) we have made considerable progress in XAI. Below we expand on our view.
> > >
> > > But overall, even if part of the AI/ML community shares our view and part shares the reviewer's view (which we respect and has forced us to think more carefully about our own position), we believe that sparking such a discussion in the community would benefit the field of XAI.
> > >
> > > Part (1):
> > >
> > > To use the reviewer's excellent aviation analogy: When pilots fly the XAI planes developed by the AI/ML community, the planes mostly don't even take off, but when they do, they tend to crash (this is clear from related work and anecdotal evidence from practice; nobody has disputed this so far). The reviewer seems to be arguing that the AI/ML community is well aware of everything, including that
> > >
> > > * the planes are not flying or are crashing,
> > > * functional evaluation metrics do not predict whether the plane will fly or crash,
> > > * there is a pressing need of involving humans in testing the planes.
> > >
> > > However, involving humans makes research more difficult, so it makes sense that we keep relying on functional evaluation, while maybe some other field tries to answer the question "What makes planes take off and fly without crashing?".
> > >
> > > If that is the case, then we would just change our position to the AI/ML community being irresponsible and jeopardizing the progress of the entire field out of sheer convenience. However, we believe that most of the AI/ML community is not aware of all of the above, hence our paper and position. In particular, we believe that top AI/ML venues set an example (and standards) that the community then follows. And currently it is good enough to publish new planes without any evidence (direct or proxy) that they might actually fly. On the other hand, if someone demonstrates the flight of an existing plane, they are likely to get rejected, because they don't contribute a novel plane design.
> > >
> > > The reviewer uses the term "laboratory conditions". This obscures the fact that our argument includes laboratory conditions testing on humans - we only argue against purely functional (without humans) evaluation. We suppose an argument could be made for purely "humanless" jet engine development, but how early in the development of flight controls or of cockpit consoles do they start involving humans? A recent review by Brighton and Klaus (Categorization of Select Cockpit Performance Evaluation Techniques) suggests that all metrics either involve humans or are proxy measures validated on humans.
> > >
> > > And we are not even arguing against the use of proxy measures in XAI. Great, if we can save time and effort. But at some point we need to establish a set of metrics that actually predict something downstream.
> > >
> > > Part (2):
> > >
> > > It is difficult to prove the absence of progress in XAI. On the other hand, if progress has been made, say, in the last 15 years, it should be relatively easy for the reviewer to find a convincing example. It is easy to confuse all the activity in XAI for progress, but when we look at the practical utility, we find little progress.
> > >
> > > The best we can do is to draw parallels with other fields of ML. Image, video, text recognition: tremendous progress in the past 15 years, many tasks that were out of reach can now be fully automated. Tabular predictive modeling: saturated field, but we can argue that there has been some progress over random forests in the last 15 years in terms of ease-of-use and inductive bias. In synthetic data generation we've come from almost nothing to generating at least tabular data that has high utility and fidelity. In XAI we have?
> > >
> > > At first glance we can say that we have SHAP and LIME. But feature contribution methods have been available since forever AND that there is no evidence that either SHAP or LIME have better practical utility that any other reasonable feature importance method. So where is the progress?
> > >
> > > Similarly in other areas of XAI (saliency, counterfactuals, prototype-based, text-based explanations) - we have 100s (if not 1000s) of methods, where each next method, if at all, is supported by better fidelity or better compactness or some other functional metric. Does the user benefit from better fidelity? We don't know. How much better compactness to achieve a practical impact? We don't know. Do feature importance method or counterfactuals have better compactness? We can't say and probably never will be able to, at least not without users (the reviewer didn't provide any further counter-argument to this inability of functional evaluation to compare methods of different types).
> > >
> > > And, to reiterate, most applications outside of model diagnostics fail. At this point it is not even clear if XAI has much value in practice outside of AI/ML expert use in model diagnostics. And, as we already said, there has been little actual progress in model diagnostics in the last 15 years.

---

### Official Review · Reviewer_ygR3 · 2025-03-14

**Significance:** 3
**Argument Clarity:** 3
**Rating:** 4
**Confidence:** 4

**Questions:**

__Questions__

__Q1.__ On page 2, you write "several authors also call for more and better user studies" and cite examples. If this call has been made for some years, what should be taken as the specific contribution of this paper?

__Q2.__ On page 6 you write "Our view that the state of user studies in XAI is relatively poor, is generally accepted and we provide further empirical evidence". Should this be taken to mean that the empirical study is the main contribution of this paper? If so, it might be useful to state it in the abstract and introduction.


__References__

Bassan, S., Amir, G., & Katz, G. (2024). Local vs. Global Interpretability: A Computational Complexity Perspective. Proceedings of the 41st International Conference on Machine Learning, 3133–3167.

Adolfi, F., Vilas, M. G., & Wareham, T. (2025). The Computational Complexity of Circuit Discovery for Inner Interpretability. The Thirteenth International Conference on Learning Representations.

Barceló, P., Monet, M., Pérez, J., & Subercaseaux, B. (2020). Model Interpretability through the lens of Computational Complexity. Advances in Neural Information Processing Systems, 33, 15487–15498.

Vilas, M. G., Adolfi, F., Poeppel, D., & Roig, G. (2024). Position: An Inner Interpretability Framework for AI Inspired by Lessons from Cognitive Neuroscience. Proceedings of the 41st International Conference on Machine Learning, 49506–49522.

**Discussion Potential:**

3

**Paper Summary:**

The authors aim to establish that (1) user studies are key to XAI, as evaluating social desiderata requires human subjects, (2) user studies are challenging as they call for more resources and skills than work without human participants, and (3) the way user studies are conducted to evaluate explainable AI is currently below the level of established standards in human-subject-based research, to the point of being detrimental to the advancement of XAI. They support this conclusion, among other things, by analyzing hundreds of XAI papers.

## Update after rebuttal

The proposed changes seem adequate, although they could be made more concrete. I will keep my score of accept.

**Position:**

Yes

**Position In Title:**

Yes

**Related Work:**

3

**Strengths And Weaknesses:**

__Strengths__

__S1.__ The paper is very well written.

__S2.__ The arguments are clear and well explained.

__S3.__ The topic and position are in need of discussion in the field and hold potential to generate awareness and improve the state of interpretability/explainability.

__S4.__ It provides a concise review.


__Weaknesses__

__W1.__ Although the 3 points the authors aim to establish first are clear, the final position is stated somewhat vaguely. For instance, the title alludes to "better" user studies, and later that we must "change our mindset" regarding user studies. Similarly, in the position statement we find "well-designed user studies should be encouraged...user studies for the sake of doing a user study...should be discouraged".

__W2.__ The abstract lacks sufficient content to understand the position and related issues. It merely states that the current state of user studies is detrimental and expresses a call for "better-designed" user studies.
For instance, you write "user studies are key to evaluating explainable AI"; you could add the reason why they are key (e.g., because evaluating social desiderata needs humans). Similarly, you write "current state of user studies is detrimental to the advancement of the field."; this could be expanded to include the features of the current state and why it makes it detrimental.
Finally, you write "we call for better-designed user studies and for greater appreciation of highquality user studies in the AI community"; you could add explanation for "better-designed" and what "higher appreciation" would mean in practice.

__W3.__ At times the paper adopts the strategy of a general review rather than a position paper, reviewing general aspects that do not necessarily relate directly to the position. See for example section 3, which quickly jumps from topic to topic without much explanation, but more generally throughout.

__W4.__ In multiple sections, it is pointed out that "more and more authors advocate for \[some aspect of the current paper's position\]". Given this state of the literature, it is sometimes unclear what is the particular contribution of the current paper.

__W5.__ It is not clear to this reviewer how the 3 bullet points on page 1 connect to the conclusion that we understand "very little" about the *value* of XAI.

__W6.__ You write "it is not surprising that there is a lack of formalism". But note there have been for a while and there are currently ongoing efforts towards formalization and reconceptualization (e.g., Bassan et al., 2024; Adolfi et al., 2025; Barcelo et al., 2020; Vilas et al., 2024).

__W7 (minor).__ The use of "better" in the title leaves the position a bit vague. It would be desirable to make it more concrete, while remaining concise, if possible.

**Support:**

3

---

> ### Author Rebuttal · Authors · 2025-03-29
>
> We thank the reviewer for the comments, they’ve been most helpful in improving the paper!
>
> **W1 and W2**: We revised the Abstract (taking into account the reviewer’s recommendations), Introduction, and Conclusion to make our position more clear and consistent. We emphasize these as the main issues:
>
> * The lack of well-designed (pre-registered, reproducible, baseline/placebo,...) user studies of high fidelity to real-world use cases (real users, real application context, downstream performance),
> * the overreliance of the AI/ML community on functional evaluation of XAI.
>
> **W3.** Section 3 is the basis for part of the criticism of existing studies. Every subsection focuses on either what should be done in an XAI study, but is mostly not (targeting end users, use of placebic explanations, investigating personalized XAI, etc.), or what shouldn’t be done, but often is (indiscriminate use of crowdsourcing, reliance on functional evaluation, etc.). We believe that a thorough review of what constitutes a good study is key for later claiming that existing studies are not that good. We now make a note of this at the start of Section 3.
>
> **W4.** The argument that supports our position has several parts (user studies are key, user studies are difficult to do, current state of user studies is poor, functional evaluation is flawed, empirical investigation) and each part has different sub-parts. For several of these sub-parts, others have already pointed something out and we put an effort into crediting every such paper that we found. We believe that this is also the main reason why certain parts read as a general review (W3).
>
> **W5.** We reduced it to two bullet points. (a) User studies are key to evaluating XAI (= understanding the value of XAI) AND (b) The current state of user studies is poor. … If we agree with (a) and (b) then the conclusion must be that we don’t understand the value of XAI, because we don’t have the only means of understanding it (= user studies). See last part of response to reviewer SwGW for details.
>
> **W6.** We added these references to Section 3, stating that there are some efforts at formalization. But we still believe that there is a lack of formalism and consensus in terminology, even if we look at papers published today, because this research hasn’t had time to proliferate.
>
> **W7 (minor).** There are so many dimensions of what constitutes a good/better user study that we didn’t find a way to be precise and remain concise. We kept the current title, but immediately in the Abstract we now state precisely what we mean (see W1).
>
> **Q1.** Most related work focuses only on one particular aspect or calls for more user studies without much support. Our work goes beyond related work by integrating all findings relevant to evaluation of XAI (placebic explanations, crowdsourcing…), we introduce things that we believe to be novel (limitations of functional evaluation, the importance of links between levels of evaluation…), and we provide empirical support for our claims about the quality of user studies. And the argument is tailored to the AI/ML community.
>
> **Q2**. The empirical study is definitely a key part of the argument supporting our position, but not the only one. However, it is relatively unique in the sense that we know of no other empirical studies that investigate the quality of user studies in XAI. We already mention the study in the Abstract and in the Introduction (at the very end).

---

> > ### Comment · Reviewer_ygR3 · 2025-04-02
> >
> > Thanks for your response. The proposed changes are, in my view, adequate. However, I cannot assess them any further as they don't seem to have been implemented in new revisions of the manuscript.

---

> > > ### Author Response · Authors · 2025-04-02
> > >
> > > Thank you for reading our rebuttal!
> > >
> > > We made the changes as we described in our rebuttal. However, as far as we are aware, we can't make update the paper here until the camera-ready version. The best we can do is to post the key changes verbatim here in the comments, if that would help.

---

### Decision · Program_Chairs · 2025-04-30

**Decision:**

Accept (poster)

**Comment:**

The paper argues that progress in explainable AI (XAI) requires better-designed user studies and that current reliance on functional evaluation methods limits understanding of explanation utility. It supports this argument with an analysis of 607 papers and outlines common methodological flaws, including lack of real users, missing baselines, and unvalidated proxies.

The paper includes a dedicated “Alternative Views” section (Section 5) that addresses potential counterarguments, such as the claim that functional evaluation is sufficient or that XAI has already made progress through tools like SHAP. Reviewer SwGW raised similar concerns, maintaining a weak reject. Other reviewers found the paper’s synthesis and analysis to be useful and agreed with its core argument.

In the rebuttal, the authors clarified that their position does not reject functional evaluation entirely but critiques its dominance in the absence of user-centered validation.